# A Study of the Metabolic Pathways Affected by Gestational Diabetes Mellitus: Comparison with Type 2 Diabetes

**DOI:** 10.3390/diagnostics12112881

**Published:** 2022-11-21

**Authors:** Loukia Spanou, Aikaterini Dimou, Christina E. Kostara, Eleni Bairaktari, Eleni Anastasiou, Vasilis Tsimihodimos

**Affiliations:** 1Department of Endocrinology and Diabetes, Hellenic Red Cross Hospital, 11526 Athens, Greece; 2Laboratory of Clinical Chemistry, Faculty of Medicine, School of Health Sciences, University of Ioannina, 45110 Ioannina, Greece; 3Director of Diabetes and Pregnancy Outpatient Department, Mitera Hospital l, 15123 Athens, Greece; 4Department of Internal Medicine, Faculty of Medicine, School of Health Sciences, University of Ioannina, 45110 Ioannina, Greece

**Keywords:** gestational diabetes mellitus (GDM), metabolomics, nuclear magnetic resonance (NMR) spectroscopy, type 2 diabetes, pathway analysis, personalized medicine, pregnancy, biomarkers, macromolecules free-serum

## Abstract

Background: Gestational diabetes mellitus (GDM) remains incompletely understood and increases the risk of developing Diabetes mellitus type 2 (DM2). Metabolomics provides insights etiology and pathogenesis of disease and discovery biomarkers for accurate detection. Nuclear magnetic resonance (NMR) spectroscopy is a key platform defining metabolic signatures in intact serum/plasma. In the present study, we used NMR-based analysis of macromolecules free-serum to accurately characterize the altered metabolic pathways of GDM and assessing their similarities to DM2. Our findings could contribute to the understanding of the pathophysiology of GDM and help in the identification of metabolomic markers of the disease. Methods: Sixty-two women with GDM matched with seventy-seven women without GDM (control group). ^1^H NMR serum spectra were acquired on an 11.7 T Bruker Avance DRX NMR spectrometer. Results: We identified 55 metabolites in both groups, 25 of which were significantly altered in the GDM group. GDM group showed elevated levels of ketone bodies, 2-hydroxybutyrate and of some metabolic intermediates of branched-chain amino acids (BCAAs) and significantly lower levels of metabolites of one-carbon metabolism, energy production, purine metabolism, certain amino acids, 3-methyl-2-oxovalerate, ornithine, 2-aminobutyrate, taurine and trimethylamine N-oxide. Conclusion: Metabolic pathways affected in GDM were beta-oxidation, ketone bodies metabolism, one-carbon metabolism, arginine and ornithine metabolism likewise in DM2, whereas BCAAs catabolism and aromatic amino acids metabolism were affected, but otherwise than in DM2.

## 1. Introduction

The prevalence of diabetes in pregnancy is increasing worldwide. Most of these cases represent patients with gestational diabetes mellitus (GDM) while the remainder includes primarily preexisting type 1 (DM1) and type 2 diabetes mellitus (DM2).

Gestational diabetes mellitus is defined as diabetes diagnosed in the second or third trimester of pregnancy that was not clearly overt prior to gestation [1]. There is a lack of international agreement regarding the diagnosis of GDM. The incidence of GDM depends on the definition of the condition and is as high as 15–20% according to the new screening criteria recommended by the International Association of Diabetes and Pregnancy Study Groups (IADPSG) [2] based on the Hyperglycemia and Adverse Pregnancy Outcomes (HAPO) [3] study.

An increasing amount of studies on GDM complications and mainly the results of the HAPO [3] study demonstrate a linear association between increased levels of fasting, 1- and 2-h plasma glucose post a 75 g oral glucose tolerance test and the risk of several significant endpoints, such as birth weight above the 90th percentile, cord blood serum C-peptide level above the 90th percentile, primary cesarean delivery, neonatal hypoglycemia, premature delivery, shoulder dystocia or birth injury, intensive neonatal care admission, hyperbilirubinemia, and preeclampsia. All this evidence changed the way GDM is viewed, diagnosed, and treated. Those results have fueled research into a more detailed characterization and clarification of the metabolic pathways that lead to GDM as well as to the investigation for possible early biomarkers of the disorder.

As a metabolic disorder, GDM shares some characteristics with obesity and DM2. It occurs when beta cells fail to provide sufficient insulin amounts or in cases of increased insulin resistance (IR) [4]. However, the pathogenesis of IR in GDM and its similarity with DM2 have not been studied in detail [5].

Metabolomics is defined as the study of metabolic processes involving the small substrates, intermediates, and products of cellular metabolism, collectively known as “the metabolome” that is influenced by both genetic and environmental factors [6]. The most highlighted and promising application of the methodology is clinical metabolomics and personalized medicine which provides essential input into the etiology and pathogenesis of disease states and contributes to the discovery of biomarkers for an early and accurate detection [7].

Nuclear magnetic resonance (NMR) spectroscopy is a key platform for the measurement of the metabolome allowing the simultaneous detection of a wide range of metabolites with high reproducibility and minimal sample preparation and is being increasingly applied to large cohort studies [6]. Since the metabolome of blood represents well the whole status of the body, NMR-based metabolomic analysis of human serum/plasma has been previously used to define metabolic signatures associated with GDM. However, in most of the studies, the use of intact serum/plasma limits the number of detectable metabolites due to the considerable overlap of their signal with the abundant proteins and other macromolecules present in blood and subsequently does not allow for the precise characterization of the metabolic pathways involved [8].

In the present study, we used NMR-based metabolic analysis of the macromolecule-free serum of GDM women to increase the number of quantified metabolites for a more accurate characterization of the altered metabolic pathways leading to GDM and to examine their similarity with those associated with DM2.

## 2. Materials and Methods

### 2.1. Subjects

The study was undertaken at the Outpatient Diabetes Unit of “Alexandra” Hospital in Athens, Greece. All subjects gave their written informed consent for inclusion before they participated in the study. The study was conducted in accordance with the Declaration of Helsinki, and the protocol was approved by the Hospital Scientific and Bioethics Committee. Eligible pregnant women who attended the Diabetes Mellitus Unit for an oral glucose tolerance test (OGTT) were enrolled. In all participants, a full medical history was recorded including race, age, gestational age, parity, history of previous GDM, family history of diabetes mellitus, medication use and smoking habits. In addition, a detailed clinical examination including weight, height, BMI, pre-pregnancy BMI and blood pressure was performed. Women with DM1, DM2 or other systematic illnesses were excluded.

Gestational Diabetes Mellitus screening was performed routinely at 26–30 weeks of gestation after overnight fasting by an OGTT. According to the IADPSG 2010 criteria [2], GDM was defined as having at least one of the following: plasma glucose levels: ≥92 mg/dL at fasting state, ≥180 mg/dL 1-h after and ≥153 mg/dL 2-h after the ingestion of 75-g of glucose.

Sixty-two Caucasian women with GDM were included. Seventy-seven healthy Caucasian pregnant women with normal glucose tolerance matched for a week of gestation, BMI, blood pressure and serum lipid levels comprised the control group. No study participant had any remarkable medical history, while all women followed, more or less, a Mediterranean style of diet.

### 2.2. Samples

Venous fasting blood samples were drawn into EDTA-containing tubes (for whole blood samples) and in Vacutainer tubes for serum samples. The serum was separated by centrifugation at 1500× *g* for 15 min and an aliquot was stored at −80 °C until NMR analysis.

### 2.3. Determination of Biochemical Parameters

The serum levels of glucose and lipid parameters were measured on an AU5400 Clinical Chemistry analyzer (Beckman, Hamburg, Germany) by standard procedures. LDL-cholesterol was calculated by the Friedewald formula. HbA1c was measured in ion exchange HPLC system (Variant II, Bio-Rad Laboratories, Hercules, CA, USA).

### 2.4. ^1^H NMR Spectroscopy of Macromolecules-Free Serum

As mentioned, serum samples contain a large amount of macromolecules (proteins and lipoproteins), which prevent the detection and quantification of low abundant and small metabolites in NMR spectroscopy. For this reason, we removed macromolecules from serum samples using the centrifugal filter devices Amicon Ultra-2 mL, 3-kDa cutoff (Merck KGaA, Darmstadt, Germany). Filter devices were first rinsed three times each with 500 μL distilled water (40 °C) followed by centrifugation (4000× *g*, 25 °C for 30 min) to remove glycerol preservatives. Then, serum samples (800 μL) were transferred to centrifugal filter tubes and centrifuged for 60 min at 4000× *g*. An aliquot of the filtrated serum samples (400 μL) was diluted with 200 µL of phosphate buffer (0.2 M Na_2_HPO_4_/0.2 M NaH_2_PO_4_, pH 7.4) and sodium 3-trimethylsilyl-(2,2,3,3-H4)-1-propionate (TSP) and transferred to 5 mm NMR tubes with a final concentration 0.456 mmol/L of TSP for the NMR measurements.

^1^H NMR spectra of the serum samples were acquired on a 11.7 T Bruker Avance DRX NMR spectrometer (NMR Center, University of Ioannina) at a proton frequency of 500.13 MHz and at a constant temperature of 300 K. A Bruker standard 1D Nuclear Overhauser Enhancement Spectroscopy (NOESY) presaturation pulse sequence RD-90°-*t*1-90°-*t*m-90°-FID, with a mixing time of 0.1 s, an acquisition time of 3.28 s and a relaxation delay of 4 s was used for all NMR experiments to suppress the water signal. For each sample, the ^1^H NMR spectrum was collected with 128 scans into 64K computer data points with a spectral width of 10.000 Hz. The free induction decays (FIDs) were multiplied with an exponential line broadening factor of 0.3 Hz prior to Fourier transformation. The phase and baseline of ^1^H NMR spectra were manually corrected by applying a simple polynomial curve fit with the Topspin software package version 4.0.6 (Bruker Biospin, Rheinstetten, Germany) and the chemical shifts were referenced to TSP (δ = 0.00 ppm). The identification of serum metabolites was based on the available databases such as the Human Metabolome Database (HMDB; http://www.hmdb.ca (accessed on 16 November 2022)), the Biological Magnetic Resonance Data Bank (BMRB, http://www.bmrb.wisc.edu (accessed on 16 November 2022)), the spectral reference libraries of Chenomx NMR Suite 8.4 (Edmonton, AB, Canada) software, J-res 2D experiments, and the existing NMR-based metabolomics literature.

### 2.5. Targeted Metabolite Profiling

For identification and quantification of the metabolites, processed NMR spectra of serum samples were imported into Chenomx NMR Suite software (version 8.4, Chenomx, Edmonton, AB, Canada). The 500 MHz spectral reference library from Chenomx and the above-mentioned TSP of known concentration as an internal standard were used for the calculation of the quantitative values of metabolites expressed in micromoles per liter (µM).

### 2.6. Statistical Analysis

Data entry and analysis were performed with SPSS software (version 23.0; IBM Corp., Armonk, NY, USA). All data are expressed as mean value ± SD. Group comparison was performed using independent samples *t*-test and *p* value < 0.05 was considered to indicate statistical significance. ROC analysis was used to test the ability of various sets of clinical and metabolic parameters to predict the presence of GDM.

### 2.7. Untargeted Metabolite Profiling

For the untargeted metabolite profiling, multivariate statistical analysis was used to investigate serum metabolic signatures of pregnant women with and without GDM. Initially, metabolite concentration data were normalized using Metaboanalyst v.4.0 and an unsupervised principal component analysis (PCA) was applied to detect clustering trends between groups and the presence of potential outliers. Then, a supervised partial least squares discriminant analysis (PLS-DA) was carried out to maximize the separation between the two groups according to their different metabolic profiles. The performance of the models was tested by a 10-fold cross validation where R^2^ and Q^2^ parameters indicate the proportion of data variance and the predictive ability of the model, respectively. Permutation tests were also carried out to check overfitting and the validation of the resulting PLS-DA models. The PLS-DA scores plot, where each point represents a sample, was used to visualize any grouping trend or separation, whereas the variable importance in projection (VIP) plot was used to highlight the most important metabolites responsible for the grouping trend or separation seen in the PLS-DA model.

## 3. Results

The anthropometric and metabolic characteristics of the study participants are summarized in Table 1. There was no difference in age, gestational week, BMI, or blood pressure between the women in the two groups. As expected, women who were classified as having GDM according to the IADPSG criteria had significantly higher glucose levels at any time of the OGTT and HbA1c levels than controls. Cholesterol, triglycerides, HDL, and LDL-cholesterol levels did not differ between the study groups.

### 3.1. Targeted Metabolite Profiling

A representative 500 MHz ^1^H NMR spectrum with the main metabolites in macromolecules-free serum from a pregnant woman with GDM, is shown in Appendix A.

Fifty-five metabolites were identified in the ^1^H NMR spectra of control and GDM groups and quantified using the Chenomx NMR Suite 8.4 software. The concentration values for each group are summarized in Table 2.

The results show that 25 of the 55 quantified metabolites were significantly altered by the presence of gestational diabetes. Compared with normal pregnancy, women who were diagnosed with GDM showed significantly elevated levels of the ketone bodies 3-hydroxybutyrate and acetoacetate, of the organic acid 2-hydroxybutyrate and the metabolic intermediates of BCAAs, 3-hydroxyisobutyrate and 3-hydroxyisovalerate. Significantly lower in GDM patients were the levels of the metabolites involved in one-carbon metabolism (methionine, glycine, serine, choline, sarcosine), in energy production (carnitine and pyruvate), purine metabolism (hypoxanthine), the amino acids arginine, asparagine, aspartate, histidine, glutamate, phenylalanine and tryptophan, the catabolic product of BCAAs 3-methyl-2-oxovalerate, the intermediate of urea cycle ornithine as well as 2-aminobutyrate and the osmolytes taurine and trimethylamine N-oxide. A model including the classical risk factors for GDM (pre-gestational BMI, number of previous pregnancies, history of GDM in previous pregnancies, parental history of type 2 diabetes) had a moderate ability to predict the presence of GDM in our population (AUC 0.659, *p* < 0.001). The inclusion of metabolic markers representative of disturbances in various metabolic pathways (3-hydroxybutyrate, 2-hydroxybutyrate and carnitine) considerably increased the accuracy of the model (AUC 0.803, *p* < 0.001).

### 3.2. Untargeted Metabolite Profiling

For the untargeted analysis, the data set consisted of the concentrations of the serum metabolites derived from the spectra of the pregnant women (62 with GDM and 77 without GDM). Prior to multivariate analysis, metabolite concentration data were normalized on Metaboanalyst v.4.0 to improve the overall consistency of the experimental data. An unsupervised (PCA) was first applied to obtain a data overview for the detection of any group trend or potential outliers (located outside the 95% confidence region of the model), (data not shown). The initial PCA score plot obtained for the two groups studied highlighted five outliers of the GDM group, each of which had a relatively elevated but not unusual concentration of one of the metabolites alanine, taurine, 2-oxoisocaproate, 2-oxocaproate and serine and thus exclusion of these samples was not considered. Then, the supervised multivariate method PLS-DA was applied and a good separation with a degree of overlap between GDM women and control was observed in the score plots of the PLS-DA model (Figure 1a). This separation is estimated by the two quality parameters, R^2^ = 0.667 for the explained variation and Q^2^ = 0.429 for the predictive capability of the resulting model (Figure 1b). In addition, permutation tests (n = 1000 repeats) were performed, and the observed statistical *p* value (*p* < 0.001) also confirms the validity of the model (Figure 1c). The VIP scores were used to identify the metabolites contributing mostly to the separation observed in the score plots of the PLS-DA model (Figure 1d). Among the top fifteen metabolites with the highest significance in the group discrimination (VIP ≥ 1), choline, phenylalanine, hypoxanthine, glycine, arginine, carnitine, tryptophan, sarcosine,3-methyl-2-oxovalerate and taurine were down-regulated in GDM women whereas 2-hydroxybutyrate, acetoacetate, 3-hydroxybutyrate, acetone, 3-hydroxyisobutyrate, valine, 3-hydroxyisovalerate, isobutyrate, citrate and threonine were up-regulated compared to controls.

The results in the untargeted analysis were mostly in line with those found in the targeted analysis. With both methods, all metabolites were altered in the same direction and most of them with similar statistical significance. The observed discrepancies were mainly attributed to the high values of SD in the targeted analysis.

## 4. Discussion

In the present study, we employed NMR-based metabolomic analysis to characterize the serum metabolic profile of pregnant women with and without GDM and to gain new insights into GDM-specific metabolic pathways. The macromolecules-free-serum samples used, allowed enriched information based on the 55 quantified metabolites, a more reliable multivariate statistical analysis and, more importantly, a deeper characterization of the affected metabolic pathways in GDM. Our results suggest that the metabolic pathways affected in GDM in a manner similar to that observed in DM2 are beta-oxidation, ketone body metabolism, aspartate/asparagine metabolism, one-carbon metabolism, arginine and ornithine metabolism. On the other hand, BCAAs catabolism, aromatic amino acids metabolism and gut microbiota were affected, but in a way different than that in DM2.

Serum levels of carnitine were found significantly reduced in GDM women, an observation consistent with the findings of similar studies [9,10]. Carnitine is an essential metabolite obtained from the diet and also biosynthesized from the amino acids lysine and methionine. It plays a crucial role in fatty acid metabolism and energy production and its deficiency results in impaired lipid metabolism [11,12,13]. The main function of carnitine in mammalian tissues is the transport of long-chain fatty acids into mitochondria as acyl-carnitine derivatives for the beta-oxidation and then for the generation of ATP through the TCA (tricarboxylic acid) cycle or ketone formation (Figure 2) [14].

Therefore, the consumption of carnitine in the translocation of acyl groups may lead to the reduction in its concentration in systemic circulation. The observed decreased plasma carnitine levels during healthy pregnancy [15,16] may reflect an increased uptake from the maternal circulation through the placenta to fuel beta-oxidation for normal fetal development [17,18]. Moreover, in late gestation, the ability of insulin to suppress lipolysis in adipose tissue is reduced resulting in increased release of non-esterified fatty acids (NEFAs) in maternal circulation and subsequently higher carnitine consumption [19], a phenomenon enhanced in GDM probably due to decreased insulin receptor substrate-1 (IRS-1) levels [20,21].

As a result, the buffering role of carnitine in increased beta-oxidation which accompanies GDM may be the main cause of the reduced carnitine levels in GDM women in our study. Apart from pregnancy, several studies revealed that *“lipid oversupply”* causes or worsens insulin resistance via multiple mechanisms such as the accumulation of intracellular lipids in many tissues [22]. More specifically, the accumulation of incompletely metabolized fatty acids in the muscle causing ‘mitochondrial stress’ inhibits both insulin signaling and glucose oxidation. In this way, agents that reduce this accumulation, such as carnitine supplementation, could be beneficial in the treatment and/or prevention of insulin resistance and DM2. Thus, the association between reduced free carnitine levels and impaired glucose homeostasis is also a common finding in DM2 patients [23].

In the mitochondria of liver cells, acetyl-CoA derived from NEFAs beta-oxidation or from pyruvate oxidation via pyruvate dehydrogenase (PDH) (Figure 2) is oxidized through the TCA cycle or is used for the ketone bodies synthesis. Oxaloacetate, a molecule crucial for the entry of acetyl-CoA into the TCA cycle, can arise from pyruvate carboxylation by pyruvate carboxylase (PC) or from transamination of aspartate (Figure 2). Low availability of oxaloacetate diminishes the capacity of acetyl-CoA to enter the TCA cycle leading to elevated ketone body production. This oxaloacetate depletion may be due to increased gluconeogenesis, a well-known feature of the GDM [9,24]. Thus, a large amount of oxaloacetate is transformed into glucose, leaving small amounts to combine with acetyl-CoA and enter the TCA cycle. This sidetrack towards ketone body production is further enhanced in a circumstance of rapid and excess production of acetyl-CoA from fatty acids resulting from reduced carbohydrate intake; prolonged, excessive alcohol consumption; and/or insulin deficiency and resistance. Thus, low availability of oxaloacetate may contribute to the statistically elevated levels of ketone bodies (3-hydroxybutyrate and acetoacetate) found in GDM women in the present study and in other metabonomic studies on GDM women at different stages of gestation [9,25,26,27,28]. In DM2, the association of ketone bodies with insulin sensitivity is conflicting. Mahendran et al. attribute this discrepancy to the presence or absence of the C allele of the GCKR gene that encodes glucokinase regulatory protein and is significantly associated with elevated levels of fasting 3-hydroxybutyrate levels [29].

In our study we observed for the first time significantly reduced levels of aspartate in the GDM group indicating an effort for oxaloacetate production (Figure 2). In addition, we found for the first time significantly lower levels of asparagine, a non-essential amino acid, which may result from the low availability of the precursors aspartate and oxaloacetate. In non-pregnant status, reduced asparagine levels were associated with the incidence of DM2 and coronary artery disease (CAD) in the Malmo Preventive Project [30] and aspartate was found among the amino acids significantly associated with reduced insulin secretion in the METSIM study [31].

One-carbon (1C) metabolism includes the folate cycle, the methionine cycle and the transsulfuration pathway (Figure 3) [32].

It is essential for many cellular processes such as methylation reactions, DNA replication, amino acids homeostasis (glycine, serine, methionine and histidine) and redox defense [33]. The cytosolic 1C pool can be fed directly by the reversible conversion of serine to glycine, where one-carbon transfer generates 5,10-methylene-THF, a molecule also formed by the histidine degradation pathway, an intermediate of folate cycle [34]. In mitochondria, 1C units can be provided also by the conversion of serine to glycine and by the glycine cleavage system [35]. Methionine is also a source of methyl groups through its conversion to *S*-adenosylmethionine (SAM) the reactive methyl carrier involved in many cellular reactions, which consequently converted to *S*-adenosylhomocysteine (SAH) [36,37]. It is reported that methionine levels regulate the ratio of SAM to SAH which impact many methylation reactions such as histone methylation [38]. In normal pregnancy, deficiencies in one-carbon metabolism are more pronounced during intrauterine development and folate deficiencies have been linked with pregnancy complications such as premature birth and neural tube defects [39,40]. Moreover, it is suggested that folic acid kinetics are non-linear and change with gestational age, but the inconsistency of these findings with other studies leaves open questions about the influence of pregnancy on one-carbon metabolism [41,42]. In the present study, one-carbon units (serine, glycine, methionine, histidine) were found significantly reduced in GDM women compared to controls (Figure 3), suggesting that one-carbon metabolism may be affected even more than in normal pregnancy. Human and animal studies have shown that maternal nutritional deficiency of one-carbon components may lead to a phenotype of diabetes and obesity in offspring [43,44]. DM2 is known to have elevated homocysteine in humans and in diabetic models and this elevation is thought to be a biomarker of impaired one-carbon metabolism. Thus, the hypothesis that impaired one-carbon metabolism may play an important role in DM2 and that correcting this impairment may represent an option for the therapeutic modulation of DM2 is now considered an exciting field of study [43].

GDM is associated with elevated oxidative stress, and consequently, GDM placenta may increase its antioxidant mechanisms to reduce oxidative damage [45], a process resulting in increased glutathione biosynthesis (Figure 3). This hypothesis may be supported by the reduced levels of glycine and glutamate in our study which are precursors for glutathione synthesis. Moreover, glycine is a gluconeogenic substrate and its reduced levels in GDM may reflect an enhanced gluconeogenesis [9]. Several studies revealed a decrease in glycine in patients with impaired insulin sensitivity, DM2 as well as in GDM pregnancies [9,46]. Glycine synthesis from serine is compartmentalized, being catalyzed by serine hydroxymethyltransferase (SHMT). SHMT in the human placenta has been pointed out as an important enzyme to cover glycine needs for fetal growth. However, alterations in the rate of conversion of serine to glycine have not been reported yet in obesity and associated metabolic disorders. A decrease in the plasma level of serine reported in those conditions may suggest a potential reduction in the activity of this pathway [46]. Disturbances in the de novo synthesis of purine due to the lack of one-carbon units [47] may be reflected by the significantly reduced levels of hypoxanthine in GDM women.

Catabolism of threonine, methionine as well as cystathionine lysis yields 2-ketobutyrate which can be further catabolized to propionyl-CoA, or 2-aminobutyrate or 2-hydroxybutyrate (2-HB) (Figure 3) [48,49]. The organic acid 2-HB is an important constituent of intermediate metabolism that may be increased in the presence of a high NADH/NAD+ ratio and demand for glutathione synthesis [48]. Recent studies have highlighted that 2-HB may be an early marker of insulin resistance and impaired glucose metabolism as well as that elevated levels in GDM may indicate a higher risk for future DM2 development [25,50,51]. In this study, in line with the findings of previous research, we found increased levels of 2-HB in the GDM group; in addition, for the first time, we report reduced levels of 2-aminobutyrate in this group. These findings that indicate the preferential formation of 2-HB rather than 2-aminobutyrate from 2-ketobutyrate, may result from: (1) a higher NADH/NAD^+^ ratio due to increased lipid oxidation and (2) elevated oxidative stress by increased circulating free radicals and/or a perturbation in antioxidant mechanisms which accompany GDM pregnancies [52].

In the present study, we found significantly reduced levels of taurine in GDM women. Taurine is an intracellular, sulphur-containing amino acid that is not incorporated into any protein but is widely distributed in all animal tissues as a free amino acid involved in many important physiological functions [53]. The level of taurine is regulated both by uptake, through the taurine transporter, and by endogenous synthesis from methionine and cysteine. Apart from a central role in osmoregulation, taurine plays an important role in the modulation of insulin secretion, acting via partially unknown mechanisms [54]. It is considered that taurine is an essential amino acid for the development of fetal islets, which contain very high levels of taurine primarily localized in the non-beta-cells and that taurine deprivation during fetal growth could lead to the development of type 2 diabetes mellitus later in life [55]. Taurine protects against the apoptosis of beta-cells and enhances their regeneration via its antioxidant and anti-inflammatory actions [56].

In the diabetic state, increased extracellular levels of glucose represent an osmotic stress for the cells and can result in cellular dysfunction [57]. Among the most important intracellular osmolytes, glycerophosphorylcholine (GPC) and sorbitol are synthesized and degraded only intracellularly, whereas specific transporter systems exist for taurine, betaine and myoinositol, which are mobile osmolytes [58]. During exposure to high extracellular levels of glucose, sorbitol is formed intracellularly via the polyol pathway, which is suspected to be one of the key processes involved in the development of diabetic complications. This intracellular accumulation of sorbitol is then most likely to cause depletion of other osmolytes such as myoinositol [59]. Given that taurine may act as a glycation scavenger preventing the intracellular formation of reactive carbonyl compounds and advanced glycation end products (AGEs) [60], such cellular dysfunction will ultimately lead to the long-term complications of diabetes in the retina, nerves, and kidneys.

Park et al. [61] reported that normal-weight GDM women had lower intakes of taurine, and that taurine intake was negatively associated with large for gestational age (LGA). They conclude that a low intake of taurine might block beta-cell expansion during pregnancy contributing to GDM and might be related to LGA development because the increase in beta-cell mass is necessary to maintain glucose homeostasis. In experimental models, pregnancy decreases organ and plasma levels of taurine, and this deficit appears to disturb fetal pancreas development. Interestingly, pups born from dams on low-protein or taurine deficient diets display glucose intolerance and when pregnant develop gestational diabetes. Furthermore, changes in taurine homeostasis have been described both in patients with DM1 and DM2 and in experimental model of diabetes mellitus [62]. The depletion in methionine, the precursor of taurine, also found in GDM women of our study further contributes to the reduced levels of this antioxidant molecule.

It is reported that the elevated levels of branched-chain amino acids (BCAAs), valine, leucine, and isoleucine, are associated with insulin resistance and may represent risk factors for type 2 diabetes [63,64]. Although in several studies the levels of BCAAs have been found to be elevated in GDM women compared to controls [65,66], these findings have not been confirmed in all circumstances [25,67]. In our study, the levels of BCAAs do not differ among the two groups per se but we observed statistically significant alterations in some metabolites related to disturbed BCAAs catabolism. 3-Hydroxyisovalerate (3-HIVA) is a byproduct of the leucine degradation pathway generated by 3-methylcrotonyl-CoA. Elevated levels of this metabolite are associated with increased ketogenesis, which is a characteristic feature of GDM. Moreover, it is reported that increased levels of 3-HIVA may result from impaired function of the enzyme methylcrotonyl-CoA carboxylase possibly due to biotin deficiency [68]. Excessive accumulation of 3-HIVA is toxic to the mitochondria and in these circumstances, carnitine is consumed for its transesterification into acylcarnitines, drained from the mitochondria and then excreted by the kidneys, resulting in low circulating levels of free carnitine [69]. 3-Hydroxyisobutyrate (3-HIB) is an intermediate of the valine degradation pathway. It is reported that 3-HIB is positively correlated with insulin resistance and increased levels of this metabolite are associated with hyperglycemia and type 2 diabetes [70]. The elevated levels of 3-HIB in the circulation and in skeletal muscles of diabetic individuals may lead to excessive free fatty acids (FFA) uptake from muscles and to defective lipid degradation, thus contributing to impaired fat oxidation and the accumulation of toxic lipid intermediates and as a consequent to insulin resistance. In previous studies, 3-HIB had a strong correlation with triglycerides, glycerol, FFA, and adipose tissue insulin resistance [71]. 3-methyl-2-oxovalerate is the branched-chain α-keto acid (BCKA) derived from isoleucine catabolism. Reduced levels of this metabolite may result from dysregulation of mitochondrial activity in GDM leading to the reduced capacity of these organelles to break down BCAAs [72]. Thus, it may be the catabolic intermediates of BCAAs that are associated with diabetes and not specifically the elevated levels of BCAAs themselves. However, it is reported that, across gestation, the branched-chain keto acids produced from the first stage of BCAAs catabolic pathway demonstrate either minimal change such as 3-methyl-2-oxobutanoic acid and 3-methyl-2-oxovalveric acid, or a decrease in concentration such as the 4-methyl-2-oxovaleric acid [73]. BCAAs and BCKAs are significantly elevated in both individuals with impaired fasting glucose (IFG) and subjects with DM2 [74]. Moreover, in a study on BCAAs catabolic pathways, among the intermediated metabolites identified, 3-methyl-2-oxovalerate was the strongest predictor of IFG after and independently of glucose [72].

Aromatic amino acids are referred to associate with insulin resistance, incident, and existing DM2 [72,75,76]. Although in several studies higher levels of aromatic amino acids especially phenylalanine and tyrosine were associated with GDM [28,77], other studies in found no obvious relationship in levels of these amino acids with gestational insulin resistance [77,78]. In our study, the levels of phenylalanine, histidine and tryptophan were found significantly reduced in GDM women compared with controls while tyrosine did not differ among the two groups. It is reported that in normal pregnancy phenylalanine decreases during pregnancy probably due to its consumption for nitrogen demands and tissue synthesis. In addition, it is reported that biochemical and metabolic changes in GDM reflect alterations in the tryptophan metabolic pathway mainly in the production of serotonin [79].

Arginine is classified as a conditionally essential amino acid, as it may be required depending on the status of the individual, such as growth during infancy, pregnancy and burn injuries [80]. In addition to being a proteinogenic amino acid, arginine is involved in the regulation of many other biological processes, such as vasodilation, calcium signaling, regeneration of adenosine triphosphate, neurotransmission, cell proliferation, and immunity [81,82] as well as the stimulation of hormones secretion, such as insulin, growth hormone, glucagon and prolactin [80]. Arginine is the unique substrate of nitric oxide synthases (NOS) for the biosynthesis of nitric oxide (NO), the endothelium-derived relaxing factor, which plays a critical role in endothelial health and vascular homeostasis [83]. In pregnancy, the demands for L-arginine are increased because of the fetal and placental growth, the increased catabolism in pregnant women and the increased production of nitric oxide (NO) for vasculature function. Diets deficient in L-arginine have been linked to poor pregnancy outcomes including fetal loss, intrauterine growth retardation, and preeclampsia [84]. In gestational diabetes, a decrease in adenosine uptake into endothelial cells results in extracellular adenosine accumulation leading to increased L-arginine transport via human cationic amino acid transporters (hCATs), increased eNOS activity, and subsequently enhanced NO synthesis. This process is the so-called adenosine/L-arginine/nitric oxide (ALANO) signaling pathway that results in endothelial dysfunction in the fetoplacental macro and microcirculation [85]. Consequently, the circulating levels of arginine in GDM could be expected to be reduced which agrees with the findings of our study. Moreover, in DM2, impaired NO bioavailability occurs early and contributes to the progression and prognosis of cardiovascular complications [86]. Moreover, the reduced levels of arginine and glutamate which are precursors for ornithine synthesis may contribute to the lower serum levels of this metabolite and may reflect a dysregulated urea cycle in GDM women.

We found statistically significant reduced levels of choline and trimethylamine-N-oxide (TMAO) and reduced but not significant levels of betaine. Choline is an essential nutrient, which serves as a precursor for the synthesis of phospholipids such as phosphatidylcholine and of the neurotransmitter acetylcholine [87]. During fetal development, these choline-derived products are essential for membrane biogenesis, cellular division, lipid transport and a reduced risk of neural tube defects [88]. Betaine, which is the oxidized product of choline, serves as a methyl donor for the formation of SAM, a crucial compound for maintaining the fetal epigenome. TMAO is a small amine oxide generated from choline, betaine, and carnitine by gut microbial metabolism. Its levels show wide inter and intra-individual variability in humans that can likely be due to multiple factors including diet, the gut microbiota, levels of the TMAO-generating liver enzyme Flavin-containing monooxygenase 3 (FMO3) and kidney function [89]. The lower levels of choline in GDM in our study is a finding consistent with experimental studies where maternal choline and betaine supplementation in GDM mice mitigates fetal overgrowth and excess adiposity [90]. However, the lower levels of TMAO are not in agreement with a study in a Chinese population that found a positive relationship between plasma TMAO concentration and GDM with a possible explanation for the discrepancy being the different microbiota in the two populations [91]. A considerable number of studies have examined the association of these metabolites with insulin resistance and DM2. Experimental models have shown that a choline-deficient diet increases liver fat content but does not cause insulin resistance in diet-induced obesity [92]. TMAO has been shown to induce DM2 via increasing fasting insulin levels and insulin resistance (HOMA-IR) in animal models, whereas studies in humans have shown that high TMAO levels are associated with an increased risk of DM2 [93]. Zeng et al. examined the association of intestinal microbiota and microbiota-generated metabolites with glucose metabolism in Chinese adults and found that plasma choline was positive, while betaine was negatively associated with diabetes, independently of TMAO levels [94].

Certain limitations of our study should be mentioned. Most importantly, the cross-sectional design of the trial does not allow the assessment of the time course of the metabolic disturbances observed in our GDM patients. So, we are unable to propose what metabolite (or set of metabolites) could help in the early diagnosis (and possibly early treatment of GDM). Likewise, the ability of the observed metabolic perturbations to predict adverse clinical outcomes or the future development of DM2 must be studied in future trials.

## Figures and Tables

**Figure 1 diagnostics-12-02881-f001:**
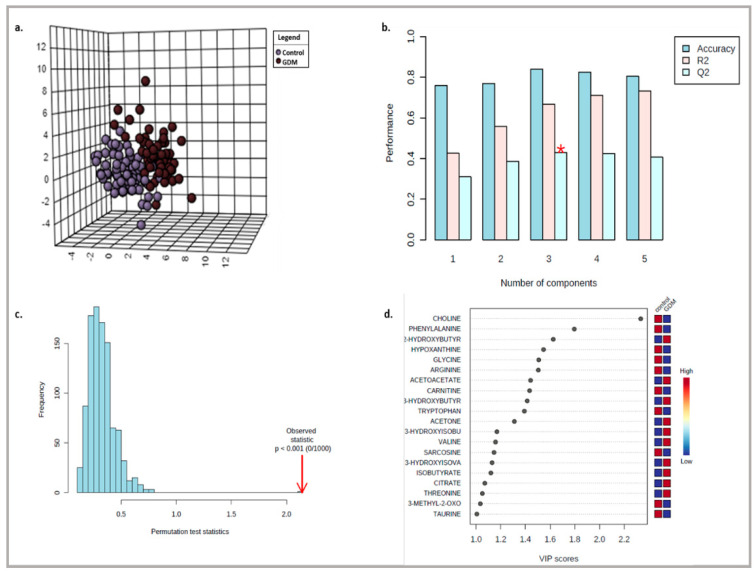
The PLS-DA multivariate analysis obtained for the 62 women with GDM and 77 healthy pregnant women: (**a**) Scores plot; (**b**) cross validation; (**c**) permutation test and; (**d**) the top 20 most discriminating metabolite GDM cases from control ranked by variable importance in projection (VIP) scores of PLS-DA model. VIP scores ≥ 1 were considered significant.

**Figure 2 diagnostics-12-02881-f002:**
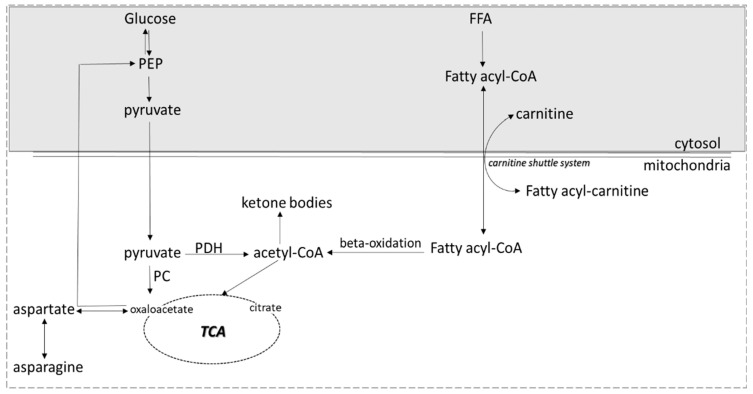
Affected metabolic pathways in GDM (pyruvate metabolism, beta-oxidation, ketone bodies metabolism).

**Figure 3 diagnostics-12-02881-f003:**
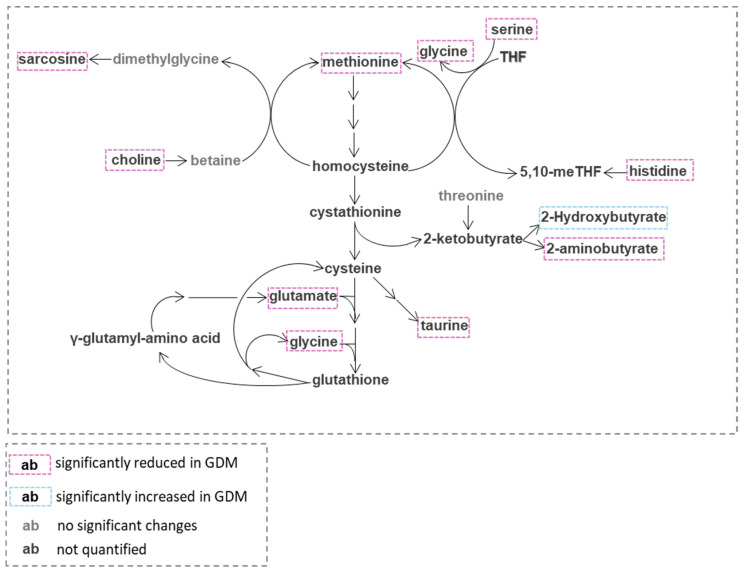
Choline, threonine and one-carbon metabolism; 5,10-meTHF:5,10-Methylenetetrahydrofolate.

**Table 1 diagnostics-12-02881-t001:** Anthropometric and metabolic characteristic of the women included in the study.

Parameter	Control Group	GDM Group	*p* Value
Ν	77	62	
Age (years)	32.69 ± 4.72	34.27 ± 5.07	NS
Gestational Week	28.06 ± 1.29	27.91 ± 1.22	NS
BMI (kg/m^2^)	25.61 ± 5.01	26 ± 4.98	NS
Systolic Blood pressure (mm Hg)	109.21 ± 13.15	111.05 ± 11.74	NS
Diastolic Blood pressure (mm Hg)	59.74 ± 7.72	59.74 ± 12.07	NS
**Metabolic Parameters**			
Total cholesterol (mg/dL)	251.08 ± 34.78	254 ± 48.45	NS
Triglycerides (mg/dL)	190.96 ± 66.85	191.15 ± 76.13	NS
HDL- cholesterol (mg/dL)	63.54 ± 12.02	64.40 ± 12.22	NS
LDL- cholesterol (mg/dL)	149 ± 32	154 ± 37.32	NS
Fasting glucose (mg/dL)	83.04 ± 5.60	93.19 ± 9.08	<0.001
Glucose 60′ (mg/dL)	130.27 ± 26.39	177.48 ± 29.90	<0.001
Glucose 120′ (mg/dL)	103.77 ± 24.31	129.26 ± 34.74	<0.001
HbA1c (%)	4.98 ± 0.29	5.13 ± 0.34	<0.01

**Table 2 diagnostics-12-02881-t002:** Concentrations of serum metabolites in the study groups.

	Metabolites (µM)	Control	GDM	*p* Value
1	2-Aminobutyrate	14.54 ± 2.55	13.28 ± 3.51	<0.05
2	2-Hydroxybutyrate	39.05 ± 14.07	49.29 ± 17.78	<0.001
3	2-Hydroxyisovalerate	8.08 ± 2.31	8.80 ± 2.94	NS
4	2-Oxocaproate	8.68 ± 2.69	8.59 ± 3.33	NS
5	2-Oxoglutarate	13.44 ± 4.28	12.74 ± 3.85	NS
6	2-Oxoisocaproate	20.62 ± 5.12	20.87 ± 6.56	NS
7	3-Hydroxybutyrate	96.93 ± 83.45	145.15 ± 124.75	<0.05
8	3-Hydroxyisobutyrate	5.54 ± 1.85	6.40 ± 2.33	<0.05
9	3-Hydroxyisovalerate	4.04 ± 1.08	4.66 ± 1.60	<0.05
10	3-Methyl-2-oxovalerate	11.79 ± 2.73	10.43 ± 2.96	<0.01
11	Acetate	46.86 ± 16.58	48.53 ± 18.55	NS
12	Acetoacetate	36.00 ± 29.43	56.79 ± 44.91	<0.01
13	Acetone	16.91 ± 9.13	18.39 ± 11.77	NS
14	Alanine	462.08 ± 84.31	442.12 ± 108.42	NS
15	Arginine	125.55 ± 13.57	109.37 ± 6.13	<0.001
16	Asparagine	58.36 ± 7.51	53.56 ± 6.70	<0.01
17	Aspartate	36.04 ± 6.27	31.40 ± 7.38	<0.001
18	Betaine	38.85 ± 9.45	37.76 ± 12.72	NS
19	Carnitine	32.14 ± 6.39	26.60 ± 6.92	<0.001
20	Choline	18.40 ± 4.29	13.07 ± 3.16	<0.001
21	Citrate	162.35 ± 35.27	172.10 ± 37.03	NS
22	Creatine	42.47 ± 12.49	40.83 ± 12.88	NS
23	Dimethylamine	4.29 ± 1.91	3.67 ± 1.96	NS
24	Formate	19.19 ± 7.58	17.13 ± 6.86	NS
25	Glutamate	98.30 ± 21.07	88.56 ± 24.95	<0.05
26	Glutamine	440.13 ± 56.78	437.82 ± 69.07	NS
27	Glycine	292.86 ± 53.47	232.30 ± 54.93	<0.001
28	Histidine	100.90 ± 24.27	86.83 ± 19.93	<0.001
29	Hypoxanthine	19.15 ± 5.67	13.87 ± 6.43	<0.001
30	Inosine	6.02 ± 0.82	5.79 ± 0.95	NS
31	Isobutyrate	9.19 ± 3.13	10.12 ± 3.28	NS
32	Isoleucine	57.26 ± 8.90	55.45 ± 10.87	NS
33	Lactate	2388.66 ± 727.39	2326.54 ± 819.48	NS
34	Leucine	120.97 ± 16.21	115.68 ± 22.86	NS
35	Lysine	170.45 ± 28.42	163.33 ± 32.25	NS
36	Malonate	14.31 ± 3.39	13.80 ± 4.43	NS
37	Mannose	120.14 ± 41.99	114.69 ± 40.03	NS
38	Methanol	52.10 ± 18.46	46.92 ± 19.88	NS
39	Methionine	28.38 ± 5.51	25.36 ± 5.61	<0.01
40	myo-Inositol	30.01 ± 8.66	30.36 ± 9.14	NS
41	N,N-Dimethylglycine	2.29 ± 0.94	2.06 ± 0.98	NS
42	O-Acetylcarnitine	4.52 ± 1.41	4.70 ± 1.57	NS
43	Ornithine	31.79 ± 10.39	26.60 ± 9.46	<0.01
44	Phenylalanine	81.21 ± 16.06	64.13 ± 14.29	<0.001
45	Proline	173.11 ± 41.93	170.37 ± 48.63	NS
46	Pyruvate	119.01 ± 39.60	105.13 ± 32.50	<0.05
47	Sarcosine	4.89 ± 1.52	3.79 ± 1.82	<0.001
48	Serine	208.51 ± 33.88	189.89 ± 34.96	<0.01
49	Succinate	7.04 ± 1.80	7.48 ± 2.36	NS
50	Taurine	141.41 ± 39.58	123.81 ± 43.30	<0.05
51	Threonine	291.61 ± 59.23	310.07 ± 79.27	NS
52	Trimethylamine N-oxide	35.45 ± 9.49	31.61 ± 9.90	<0.05
53	Tryptophan	50.31 ± 16.18	38.72 ± 16.55	<0.001
54	Tyrosine	50.85 ± 8.26	47.39 ± 12.06	NS
55	Valine	193.94 ± 27.28	198.05 ± 37.34	NS

## Data Availability

This study did not report any data.

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
