# Peer review of "A Study of the Metabolic Pathways Affected by Gestational Diabetes Mellitus: Comparison with Type 2 Diabetes"

_diagnostics, 2022, doi:10.3390/diagnostics12112881_

Round 1

Reviewer 1 Report

Diagnostics-2019981 Evaluation

This is a very interesting study that compares the metabolic pathways affected by gestational diabetes with the pathways affected by type 2 diabetes.

The manuscript is very well written, complying with all the methodological aspects, it only requires some small changes to consider its publication.

On the other hand, he considered it important that the discussion mention whether there is scientific information regarding the differences in these metabolic pathways according to the racial group to which the pregnant woman with and without gestational diabetes belongs, the type of diet, and the body mass index ( BMI) with which they start the pregnancy. 

In the summary section

It is important that you mention in the abstract what the purpose of your study was 

In the material and methods section

How were the patients selected?

What type of sampling was done?

How was the sample size calculated?

They consider whether the sample of patients studied represents pregnant women with gestational diabetes.

Line 108

What was the volume of blood obtained and at what time?

Line 158

It is necessary to place the complete information including the company and city.

Line 274

What does TCA mean? The cycle of tricarboxylic acids

The manuscript requires a list of abbreviations

At the end of the discussion section

What would be the perspectives of the study?

Reviewer 2 Report

The manuscript entitled “A study of the metabolic pathways affected by gestational diabetes mellitus. Comparison with type 2 diabetes” submitted by Spanou et al. evaluated the role of metabolic pathways in affecting gestational diabetes mellitus (GDM). The author evaluated 55 metabolites in 62 GDM cases and 77 pregnant controls and found a statistically significant difference of 25 metabolites between the two groups. My recommendation regarding the manuscript are as follows.

1.      In the subject subsection under Materials and Methods, the authors mentioned that GDM is defined as: plasma glucose levels: ≥ 92mg/dl at fasting state, ≥  180  mg/dl 1-h after and ≥  153  mg/dl 2-h after the ingestion of           75-g of glucose. But according to Table 1, the sugar levels at Fasting and after 1 and 2 hrs of OGTT of  GAD cases were below the authors' claimed values. Were authors’ claim patients have diabetes? Please clarify.

2.      No inclusion criteria of HbA1c or HbA1c-based GAM range were provided in the manuscript. The values of HbA1c in cases have seemed they may be nondiabetic.

3.      The sample size is very small. Did the authors calculate the sample size for getting a standard statistical power?

4.      4. Some grammatical errors were found that should be corrected.
